# CriSNPr, a single interface for the curated and de novo design of gRNAs for CRISPR diagnostics using diverse Cas systems

Asgar H Ansari[1,2†], Manoj Kumar[1,2†], Sajal Sarkar[1,2], Souvik Maiti[1,2], Debojyoti Chakraborty[1,2]*

[1]CSIR-Institute of Genomics & Integrative Biology, New Delhi, India; [2]Academy of Scientific & Innovative Research (AcSIR), Ghaziabad, India

**Abstract** CRISPR-based diagnostics (CRISPRDx) have improved clinical decision-making, especially during the COVID-19 pandemic, by detecting nucleic acids and identifying variants. This has been accelerated by the discovery of new and engineered CRISPR effectors, which have expanded the portfolio of diagnostic applications to include a broad range of pathogenic and non-pathogenic conditions. However, each diagnostic CRISPR pipeline necessitates customized detection schemes based on the fundamental principles of the Cas protein used, its guide RNA (gRNA) design parameters, and the assay readout. This is especially relevant for variant detection, a low-cost alternative to sequencing-based approaches for which no in silico pipeline for the ready-to-use design of CRISPRDx currently exists. In this manuscript, we fill this lacuna using a unified web server, CriSNPr (CRISPR-based SNP recognition), which provides the user with the opportunity to de novo design gRNAs based on six CRISPRDx proteins of choice (*Fn/enFn*Cas9, *Lw*Cas13a, *Lb*Cas12a, *Aa*Cas12b, and Cas14a) and query for ready-to-use oligonucleotide sequences for validation on relevant samples. Furthermore, we provide a database of curated pre-designed gRNAs as well as target/off-target for all human and SARS-CoV-2 variants reported thus far. CriSNPr has been validated on multiple Cas proteins, demonstrating its broad and immediate applicability across multiple detection platforms. CriSNPr can be found at http://crisnpr.igib.res.in/.

**\*For correspondence:**
debojyoti.chakraborty@igib.in

† These authors contributed equally

**Competing interest:** The authors declare that no competing interests exist.

## Editor's evaluation

The web-based software developed in this study will be of interest to researchers who develop CRISPR-based diagnostic methods. The use of CRISPR-Cas to rapidly identify specific mutations in both cancer and infection is an evolving field with good potential to play a role in future research and diagnostics. This software will facilitate the implementation of such technologies and is therefore useful.

## Introduction

CRISPR proteins' highly specific recognition of DNA and RNA has made them useful not only as primary gene editors but also for rapid molecular diagnosis of pathogenic nucleic acid mutations. Traditional probe-based diagnostic tests rely on a polynucleotide hybridizing to the target DNA/RNA and providing a readout via an amplification reaction to provide a readout (*Wetmur, 1991*; *Carpenter et al., 1993*; *Zhao et al., 2015*; *Thaxton et al., 2006*). These quantitative RT-PCR (qRT-PCR) tests and their derivatives have proven to be the gold standard for detecting trace amounts of pathogenic nucleic acids in samples, and they have been used globally to detect SARS-CoV-2 during the ongoing COVID-19 pandemic. Although qRT-PCR is highly sensitive to detect only a few copies of

the target pathogenic sequence, its ability to differentiate very closely related sequences has not yet been successfully demonstrated for accurate clinical diagnosis.

The association of Cas proteins with a target nucleic acid is followed by a secondary readout, either directly through the bound ternary complex or catalytic cleavage of the substrate followed by collateral cleavage of reporter molecules (*Jinek et al., 2012*; *Jinek et al., 2013*; *Sternberg et al., 2014*; *Cong et al., 2013*; *Ran et al., 2013*; *Mali et al., 2013*). DNA/RNA interrogation begins with a guide RNA (gRNA) binding to a target sequence and is then followed by the catalytic activity of Cas effectors. It has been demonstrated that for some Cas proteins, this two-step process generates a very high specificity of target recognition, which can be extended to the diagnosis of single nucleotide variants (SNVs; *Chen et al., 2020b*; *Teng et al., 2018*; *Karvelis et al., 2020*; *Li et al., 2018*; *Myhrvold et al., 2018*; *Teng et al., 2019*; *Azhar et al., 2021*; *Harrington et al., 2018*; *Acharya et al., 2019*; *Kumar et al., 2021*). In comparison to the gold standard SNV detection technologies based on Sanger/Deep sequencing, which necessitate dedicated infrastructure, manpower, and analysis pipelines, as well as longer turnaround times, CRISPR-based variant calling is an appealing alternative for rapid, low-cost diagnosis of disease-causing mutations. Currently (until January 2022), the ClinVar database contains information on approximately 117,437 (GRCh38) pathogenic human variants associated with diseases, the vast majority of which can be detected using CRISPR-based tests (*Teng et al., 2019*; *Azhar et al., 2021*; *Harrington et al., 2018*; *Kumar et al., 2021*; *Landrum et al., 2020*; *Landrum et al., 2018*; *Kellner et al., 2019*). Similarly, the rapidly evolving SARS-CoV-2 variants emphasize the importance of detecting mutations in pathogenic sequences in order to develop public health strategies, effective vaccines, and a better understanding of disease pathophysiology.

CRISPR-based diagnostics (CRISPRDx) is a relatively new addition to the arsenal of diagnostic methodologies for detecting SNVs. The majority of these pipelines rely on the ability of the Cas protein to differentiate nucleic acids based on mismatches in the gRNA at predetermined positions. This nucleotide position-specific mismatch sensitivity, first reported with *Lw*Cas13a, has now been linked to several other Cas effectors and used for SNV detection, including *Fn*/en*Fn*Cas9, *Lb*Cas12a, *Aa*Cas12b, and Cas14a (*Li et al., 2018*; *Myhrvold et al., 2018*; *Teng et al., 2019*; *Azhar et al., 2021*; *Harrington et al., 2018*; *Acharya et al., 2019*; *Kumar et al., 2021*; *Acharya et al., 2021*). Several CRISPR/Cas systems have also demonstrated protospacer adjacent motif (PAM) mismatch sensitivity, but because PAM is not always present at the target DNA/RNA sequences, their applicability for diagnostic assays is limited (*Teng et al., 2019*; *Kim et al., 2020*).

Although the overall strategy for mismatch identification remains largely consistent across Cas proteins, each Cas protein possesses distinctive properties with regard to mismatch-sensitive positions in the gRNA. These were discovered through extensive nucleic acid: protein structural and biochemical research. As a result, while the same SNV can be targeted by multiple Cas proteins, each diagnostic strategy necessitates a unique crRNA design (*Li et al., 2018*; *Myhrvold et al., 2018*; *Teng et al., 2019*; *Azhar et al., 2021*; *Harrington et al., 2018*; *Acharya et al., 2019*; *Kumar et al., 2021*; *Acharya et al., 2021*). Because each Cas system differs in terms of gRNA sequence, readout mode, PAM requirement, and mismatch sensitivity positions, it can be time-consuming for a user to first identify which Cas effector to use and then design appropriate gRNA and primers for performing diagnostic assays. Consequently, it is necessary to develop a unified method that equips any user with the minimal knowledge and information necessary for designing detection assays for any SNV of interest.

In response, we present a web server called CRISPR-based SNP recognition (CriSNPr) for designing CRISPRDx pipelines across the CRISPR platforms reported so far for variant detection. CriSNPr is a pipeline for CRISPR-based detection of pathogenic and non-pathogenic mutations in all reported human nucleotide variants (SNP Database [dbSNP]) and SARS-CoV-2 variants of interest/concern (VOI/VOC). Furthermore, it allows for the design and implementation of de novo variants of choice. The server searches for the SNV of interest and returns information about all Cas systems that can be used to detect that SNV, as well as the required crRNA and primer design parameters based on gRNA design principles available in the literature for each Cas protein (*Li et al., 2018*; *Myhrvold et al., 2018*; *Teng et al., 2019*; *Azhar et al., 2021*; *Harrington et al., 2018*; *Acharya et al., 2019*; *Kumar et al., 2021*; *Acharya et al., 2021*). Importantly, CriSNPr, unlike other available sgRNA design tools, also provides information about off-targets for SNVs targeting modified crRNA sequences and scores them based on the number of off-targets they produce.

CriSNPr has integrated mismatch-sensitive position data for *Fn*/en*Fn*Cas9, *Lw*Cas13a, *Lb*Cas12a, *Aa*Cas12b, and Cas14a and has been experimentally validated on SNVs for a subset of Cas proteins (*Li et al., 2018*; *Myhrvold et al., 2018*; *Teng et al., 2019*; *Azhar et al., 2021*; *Harrington et al., 2018*; *Acharya et al., 2019*; *Kumar et al., 2021*; *Acharya et al., 2021*). To expand its immediate application to existing human and SARS-CoV-2 variants, the current version of CriSNPr contains SNV information from the human dbSNP and SARS-CoV-2 CNCB-NGDC databases. Even without prior sequence information about an SNV, a user can avail designed sequences by using the rsID (Reference SNP cluster ID) for humans or mutant amino acid position for SARS-CoV-2 (*Sherry et al., 2001*; *Xue et al., 2021*). For new SNV-containing sequences not in CriSNPr's database, users can fetch crRNA and primer sequences by providing a sequence length of 20–30 nucleotides with an SNV position and variant nucleobase identity. Finally, we were able to experimentally detect the SARS-CoV-2 E484K variant as well as clinically relevant human SNPs rs2073874 and rs138739292 using *Fn*Cas9, Cas14a, and *Aa*Cas12b, exhibiting the efficacy of the crRNA designed by CriSNPr.

## Results
### Conventional gRNA design tools are not tailored for CRISPR diagnostic pipelines

While there are numerous in silico pipelines for gRNA design for individual Cas proteins, these are primarily intended for gene editing applications (*Haeussler et al., 2016*; *Chuai et al., 2018*; *Montague et al., 2014*; *Labun et al., 2019*; *Heigwer et al., 2014*; *Moreno-Mateos et al., 2015*; *Doench et al., 2016*; *Wang et al., 2019*; *Concordet and Haeussler, 2018*; *Kim et al., 2018*). For a given Cas9, the workflow for gRNA design integrates specificity and sensitivity scores generated

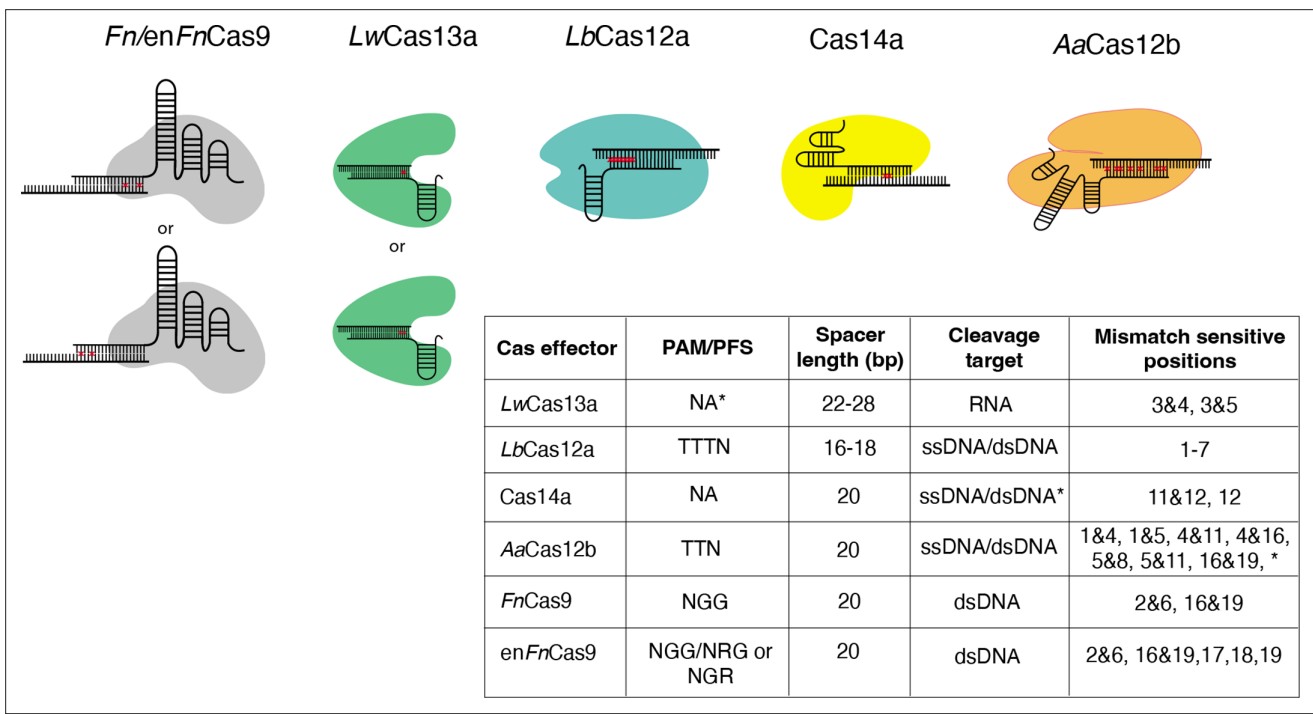

| Cas effector | PAM/PFS | Spacer length (bp) | Cleavage target | Mismatch sensitive positions |
|---|---|---|---|---|
| *Lw*Cas13a | NA* | 22-28 | RNA | 3&4, 3&5 |
| *Lb*Cas12a | TTTN | 16-18 | ssDNA/dsDNA | 1-7 |
| Cas14a | NA | 20 | ssDNA/dsDNA* | 11&12, 12 |
| *Aa*Cas12b | TTN | 20 | ssDNA/dsDNA | 1&4, 1&5, 4&11, 4&16, 5&8, 5&11, 16&19, * |
| *Fn*Cas9 | NGG | 20 | dsDNA | 2&6, 16&19 |
| en*Fn*Cas9 | NGG/NRG or NGR | 20 | dsDNA | 2&6, 16&19,17,18,19 |

**Figure 1.** Nucleotide mismatch sensitive crRNA positions have been reported for a variety of Cas systems. These positions are included in CRISPR-based SNP recognition (CriSNPr) for *Lw*Cas13a, *Lb*Cas12a, Cas14a, *Aa*Cas12b, *Fn*Cas9, and enFnCas9, respectively. A table summarizing the protospacer adjacent motif (PAM)/protospacer flanking site (PFS) for each Cas protein is shown, along with mismatched sensitive positions reported in the literature. *LwCas13a does not require PFS when targeting the mammalian genome. *Cas14a cleaves ssDNA without PAM, but dsDNA requires TTTA PAM. *AaCas12b has shown mismatch sensitivity for some other nucleotide positions as well, but they are not included here because the discrimination between wild-type (WT) and mutant is insufficient.

The online version of this article includes the following source data for figure 1:

**Source data 1.** A table summarizing various Cas proteins previously reported to have mismatched sensitive crRNA nucleotide positions.

from multiple factors, such as identification of available PAM sites relevant to the target, presence of preferred nucleotides in proximity to PAM, their context within the full-length gRNA sequence, GC (guanine-cytosine) content, and the overall correlation between the gRNA sequence and experimentally validated editing rates (*Haeussler et al., 2016*; *Chuai et al., 2018*; *Montague et al., 2014*; *Heigwer et al., 2014*; *Doench et al., 2016*). In addition to gRNA sequence features, local genetic and epigenetic characteristics, the presence or absence of structural motifs, nucleosome positioning, and other factors are taken into account for optimal gRNA design (*Chuai et al., 2018*; *Montague et al., 2014*; *Labun et al., 2019*; *Wang et al., 2019*). Specificity scores are assigned to gRNAs either through alignment to the genome or through hypothesis-driven approaches that incorporate gRNA structural information in addition to sequence data. Machine learning has recently contributed to optimal computational design parameters that empirical sequence and structure-driven gRNA prediction algorithms are unable to incorporate (*Chuai et al., 2018*; *Wang et al., 2019*; *Kim et al., 2018*).

Importantly, these algorithms are designed to make predictions about the best gene-targeting sgRNAs based on the biological parameters of DNA recognition by the individual Cas systems. In the case of CRISPR diagnostics, the paradigm has been empirically shifted; the target sgRNA binding region is relatively fixed, and diagnostic sgRNAs, corresponding amplicons, or reporters must be tailored around the target region. For each Cas used in the diagnosis of a particular SNV, the available PAMs and the guidelines for mismatch sensitivity positions are unique. Also distinct is the off-targeting propensity of gRNAs designed based on the Cas effector of choice, as shown in *Figure 1*.

Mismatch sensitivity associated with RNA targeting *Lw*Cas13a, for example, is achieved by combining an SNV with a synthetic mutation at the third and fourth or third and fifth PAM proximal nucleotide positions within a crRNA sequence (*Myhrvold et al., 2018*). According to recent findings, 5′ protospacer flanking sites are not required for *Lw*Cas13a when targeting mammalian sequences, but 'H' (A/C/T) at 5′ sites may improve target binding or cleavage (*Myhrvold et al., 2018*; *Abudayyeh et al., 2017*). *Lb*Cas12a, unlike *Lw*Cas13a, requires a TTTN PAM sequence and exhibits sensitivity at multiple single nucleotide positions from the first base to the seventh base proximal to the PAM (*Chen et al., 2020b*; *Li et al., 2018*; *Tóth et al., 2020*). As previously reported, the mismatch is sensitive when placed anywhere between the first and seventh base positions when using ssDNA or dsDNA as cleavage targets. This difference in signals between wild-type (WT) and mutated sequences can be enhanced by using shorter crRNAs of 16–18 nucleotides in length (*Li et al., 2018*). Cas14a, like *Lb*Cas12a, can cleave ssDNA as well as dsDNA targets but only when targeting dsDNA. Cas14a necessitates a T-rich 5′ TTTA PAM, whereas ssDNA does not have this requirement. Even though Cas14a has sensitivity for only 11 and 12 or 12 crRNA positions, this PAM flexibility is a proposed advantage when used for SNV detection (*Karvelis et al., 2020*; *Harrington et al., 2018*). Although *Aa*Cas12b requires 5′ TTN PAM to cleave the ssDNA/dsDNA sequences, it exhibits dual mismatch sensitivity at several positions, expanding the possible target SNV combinations (*Teng et al., 2018*; *Teng et al., 2019*). *Aa*Cas12b's dual mismatch sensitivity has been demonstrated at a wide range of crRNA positions; however, only combinations demonstrating efficient WT and mutated sequence discrimination were chosen. These include positions 1 and 4, positions 1 and 5, positions 4 and 11, positions 5 and 16, and positions 5 and 19 (*Teng et al., 2019*). Similarly, previous research suggested that *Fn*/en*Fn*Cas9 systems with dual mismatch sensitivity require mismatches at specific positions, such as 2 and 6 or 16 and 19, preventing target dsDNA binding for variant detection (*Azhar et al., 2021*; *Acharya et al., 2019*; *Kumar et al., 2021*). In addition to dual mismatch sensitivity, it has been reported that enFnCas9 also exhibits single mismatch sensitivity at the PAM distal 17th, 18th, and 19th positions (*Acharya et al., 2021*). The ability to faithfully discriminate between sequences contingent on single nucleotide mutations across the crRNA sequence provides an opportunity to repurpose and utilize all of these Cas effectors.

It is important to note that we have added *Lw*Cas13a to our web server because it has been reported for SNV detection. *Lw*Cas13a belongs to a class of RNA-targeting Cas effectors that are not constrained by PAM. As a result, their targetability across the genome cannot be directly compared to other PAM-dependent DNA-targeting Cas effectors. However, designing diagnostic assays with *Lw*Cas13a requires an additional step of converting DNA to RNA via in vitro transcription (*Myhrvold et al., 2018*).

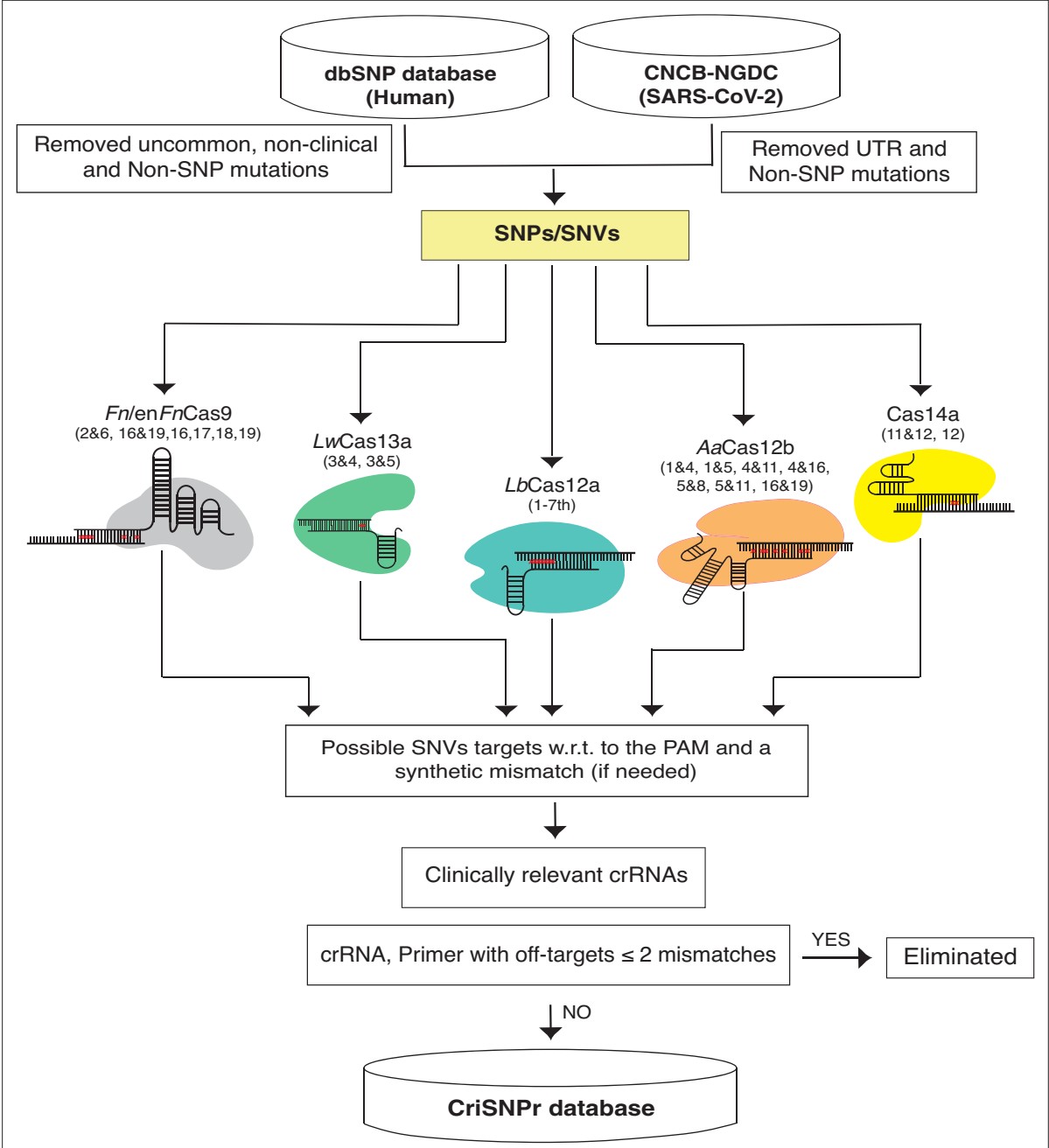

**Figure 2.** Schematic representation of CRISPR-based SNP recognition (CriSNPr) database curation and the clinically relevant SNP Database (dbSNP) variations were filtered for uncommon and non-SNP mutations and SARS-CoV-2 in non-UTR single nucleotide variants (SNVs). The filtered SNPs were then checked for targetability by individual Cas systems based on mismatch sensitivity with or without protospacer adjacent motif (PAM). The genome coordinates of target SNPs aided in the acquisition of gene IDs as well as SNP flanking sequences for oligo synthesis, resulting in the creation of an SQLite database. The off-targets were evaluated against representative bacteria, viruses, the human genome, and the transcriptome.

## CriSNPr generates readouts by querying variants in publicly available human and SARS-CoV-2 datasets

We chose NCBI's dbSNP as a reliable source of clinically relevant SNPs from one of the largest human variation databases (*Sherry et al., 2001*). This includes single nucleotide changes, insertions, deletions, and microsatellites, as well as population-level frequency, publication, genomic annotation of common variations, and pathological mutations. To begin, we isolated pathological SNPs from the most recent dbSNP Build 155 (*Figure 2*, Materials and methods) by omitting relatively uncommon,

non-clinical, and non-SNP variants. For SARS-CoV2, we filtered SNVs by removing UTR and non-SNP mutations from the most recent CNCB-NGDC SARS-CoV-2 variations database (*Xue et al., 2021*). For target SNVs/SNPs, crRNAs were designed by retaining variant bases at designated positions and then introducing synthetic mismatches at positions pertinent to experimentally validated data obtained from the various CRISPR systems, as shown in *Figure 2*. Following that, the genome coordinates of these crRNA sequences were used to obtain flanking sequences and, finally, to design primer sequences for PCR amplicons that can be used in either binding or cleavage-based reporter output assays, depending on the CRISPR proteins of choice (*Untergasser et al., 2012*). To reduce non-specific amplification by primer sequences, they were screened for off-targets on a representative bacterial genome database (NCBI), virus genome database (NCBI), and human genome/transcriptome (GENCODE GRCh38) with up to two mismatches (*Frankish et al., 2019*; *Tatusova et al., 2014*; *Brister et al., 2015*). Due to the size of dbSNP, retrieving all of this information about crRNAs and flanking primer sequences in real time can be time-consuming. To reduce this lag, the extracted information about clinically relevant SNPs, crRNAs, gene IDs, and primer sequences is formatted as an SQLite database to support the web server, as shown in *Figure 2*.

## CriSNPr provides a search-and-select pipeline for designing a CRISPRDx assay

CriSNPr's interface has three subdomains: human, with ready-made target clinically relevant SNPs from the human dbSNP; SARS-CoV-2, with ready-made target SNVs within the SARS-CoV-2 genome; and Seq-CriSNPr, for a sequence-based search of any reported target SNV in the human or SARS-CoV-2 genomes. The CriSNPr web server attempts to validate the entered input by mapping and visualizing SNVs/SNPs for related information such as organism of origin, genome coordinates, gene ID, disease association, reported allele, and population-specific frequency distribution. The ability to target a queried SNP/SNV by all included Cas systems is checked simultaneously, and only those with positive results are shown with the required information.

The interface for human variants in the CriSNPr web server takes SNP rsID as input and looks for available matching sequences in the CriSNPr database of target SNPs, returning SNP-related allelic information, crRNAs, and primer oligos for SNP detection by the Cas systems integrated into the CriSNPr, as shown in *Figure 3* and Materials and methods. The SARS-CoV-2 interface takes variant amino acid positions (e.g. S N501Y, S E484K, etc.) as input, and after finding a match in the CriSNPr database of target SNVs, it returns SNV identity information along with variation frequencies, crRNAs, and primer oligos for SNV detection in variant SARS-CoV-2 lineages by the Cas systems integrated into the CriSNPr (*Figure 3*; Materials and methods).

Since the position of mismatches differs between different Cas proteins used for designing the diagnostic pipeline, CriSNPr also returns information highlighting the position of mutant (red color) and synthetic nucleotide (needed depending on the Cas systems used, green color) within a crRNA sequence, as well as the genome coordinates of the target, the DNA strand corresponding to the crRNA of interest, and the off-targets of modified crRNA sequences. CriSNPr employs the previously reported offline versatile algorithm Cas-OFFinder to identify potential off-targets against the modified crRNAs (*Bae et al., 2014*). Using this information, CriSNPr provides off-target information against a crRNA, including chromosome location and coordinates of the off-targets, DNA strand information, and a number of off-target sequences with up to four mismatches, etc., as shown in *Figure 3*. All of the primer sequences provided by CriSNPr have been pre-filtered for off-targets with up to two nucleotide mismatches against representative bacterial genome databases (NCBI), virus genome databases (NCBI), and human genome/transcriptome databases (GENCODE GRCh38). This is especially important when the sample contains both human and pathogenic nucleic acids (such as in the SARS-CoV-2 infection). In addition, for Cas proteins without trans-cleavage activity (such as *Fn*Cas9/en*Fn*Cas9), in vitro cleavage-based discrimination is facilitated by designing primers for longer amplicons (*Figure 3*). This is especially important when considering SNV detection with clinical relevance or diagnosis, as non-specific amplicons can be generated from different species of RNA contamination. These can be resolved on an agarose gel following a CRISPR-mediated cleavage (*Azhar et al., 2021*; *Kumar et al., 2021*).

Given that the first two interfaces provide information that has been curated within the CriSNPr database, it is possible to generate the outputs in less than a minute. To target any previously unreported

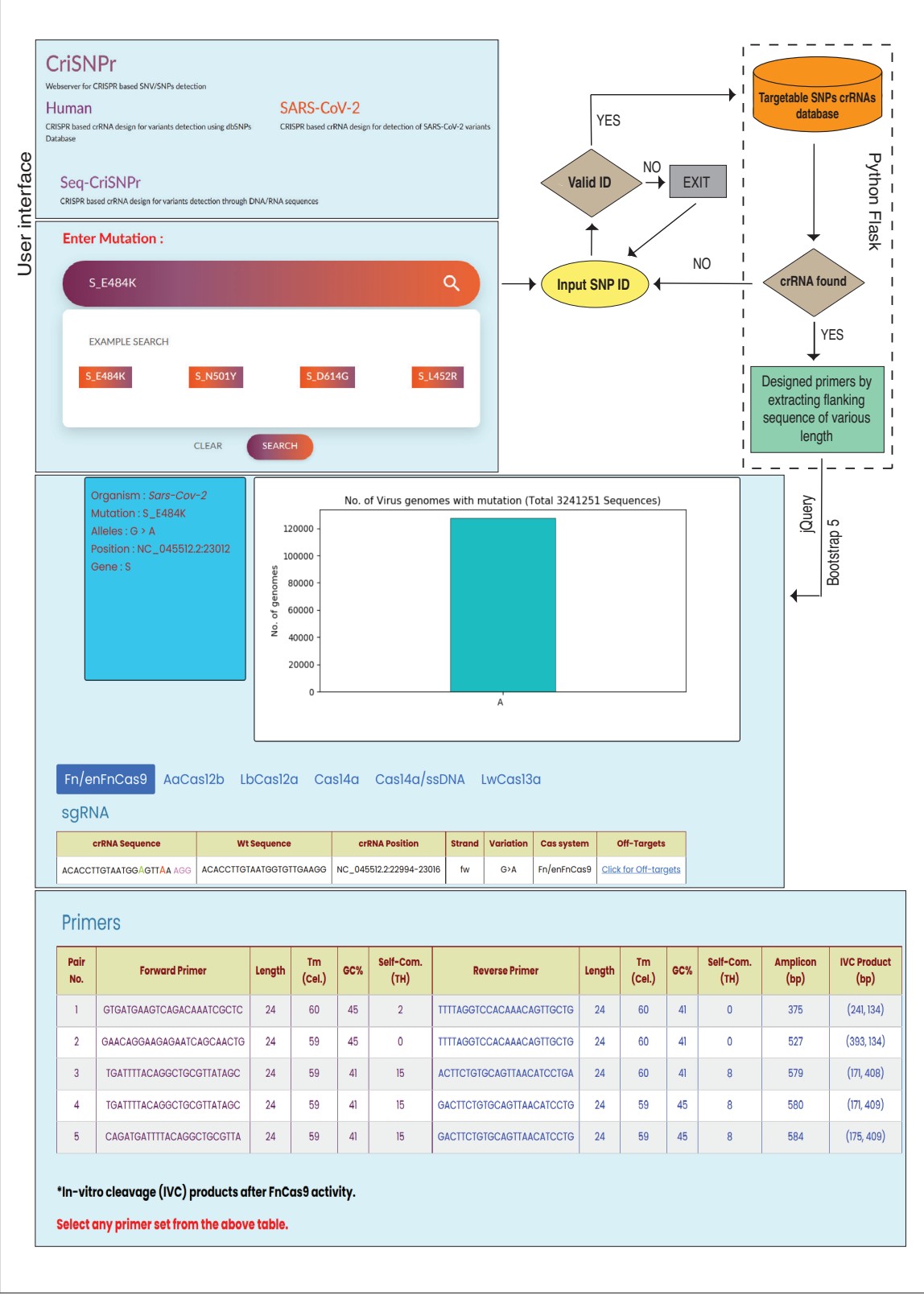

**Figure 3.** Workflow of the CRISPR-based SNP recognition (CriSNPr) web server. CriSNPr user interface displays human, SARS-CoV-2, and Seq-CriSNPr subdomains, each of which accepts rsID, mutant amino acid position, and single nucleotide variant (SNV) containing 20–30 nt sequences as inputs. With a valid input, the server will look for matching crRNA sequences in the database created with the Python Flask framework. The results include the sequences of the amplification and SNV-detection primers, the allelic distribution of the SNVs, the crRNAs, and the off-targets.

*Figure 3 continued on next page*

Figure 3 continued

The online version of this article includes the following figure supplement(s) for figure 3:

**Figure supplement 1.** Seq-CRISPR-based SNP recognition (CriSNPr) allows for the entry of any 20–30 nt.

or novel SNP or SNV present in humans or SARS-CoV-2, Seq-CriSNPr considers a 20–30-nucleotide sequence containing the SNP/SNV along with the position and nucleobase identity of the SNP/SNV as input and performs a real-time design of crRNA and primer oligos for each CRISPR/Cas detection assay, as shown in *Figure 3—figure supplement 1* and Materials and methods. When compared to the extraction of already existing information about clinically relevant SNPs/SNVs from the CriSNPr database, this process can take a few seconds extra (30–40 s). Using the same Python flask framework as CriSNPr, Seq-CriSNPr gives all the sequence information similar to CriSNPr but with more user-customizable options to choose a position as well as the identity of the SNV nucleotide within a 20–30 nt query sequence (*Figure 3—figure supplement 1*). Seq-CriSNPr currently accepts query sequences only from the human and SARS-CoV-2 genomes. To expand this to other genome sequences based on the user's need, we have offered the possibility of submitting a request via an online form.

## Different Cas systems' ability to target the dbSNP and SARS-CoV-2 genomes

As described previously, CRISPR systems demonstrating mismatch sensitivity offered diverse SNP-targeting positions. To determine the total target SNPs/SNVs within the dbSNP by any of the individual Cas effectors, we evaluated possible constraints for these systems, as a systematic comparison between different platforms for SNV targeting has not yet been reported. Among the Cas proteins included, *Lw*Cas13a (targeting RNA) and Cas14a (ssDNA) can naturally target almost all the variations in the dbSNP because neither is limited by PAM requirements (*Figure 4a*). However, *Lw*Cas13a needs an extra step to change a DNA substrate into RNA before the CRISPR reaction and readout.

Among the Cas systems with PAM requirements, the enhanced version of *Fn*Cas9 (en*Fn*Cas9) had the highest number of target SNVs (85.15%, 15,988,454 SNPs). This is followed by *Aa*Cas12b, *Fn*Cas9, and *Lb*Cas12a with target SNPs of 62.32% (11,702,008), 43.86% (8,234,812), and 29.95% (5,623,279), respectively. Cas14a could target 500,786 variations (2.66%) of the total 18,775,119 variations when the substrate is double-stranded target DNA due to the TTTA PAM constraint. These findings imply that, while the number of sensitive mismatch positions increases the chances of targeting an SNP from dbSNP, PAM relaxation provides more coverage for detecting SNVs. *Lb*Cas12a, on the other hand, despite seven single mismatch sensitive positions, targets only 5,623,279 SNPs due to a stringent TTTN PAM. In comparison, *Aa*Cas12b detects more SNPs (62.32%) due to relaxed TTN PAM and various combinations of dual mismatch sites. But since Cas14a with dsDNA targets is only sensitive for two positions and uses a very stringent TTTA PAM, the overall detection ratio is the lowest of any PAM-dependent SNP detection Cas system, as shown in *Figure 4a*.

Following that, we examined the potential disease-causing/associated SNPs detected by various Cas systems. A total of 493,105 disease-related SNPs were extracted from the SNPs that were found in dbSNP after they were filtered. Among PAM-dependent Cas systems, en*Fn*Cas9 and Cas14a (dsDNA) exhibit the highest (90.74%) and lowest (1.50%) SNP detection rates, respectively, as depicted in *Figure 4b*. Interestingly, aside from *Fn*/en*Fn*Cas9, all of the other PAM-dependent Cas systems showed a lower number of disease-related SNPs that can be targeted. This could be due to the presence of NRG/NGR PAMs near the majority of these disease-related SNPs, as shown in *Figure 4b*. This could be further explained by examining the distribution of different mutation classes among SNVs of interest. Total and clinically relevant target mutations in the dbSNP are enriched for the G.C > A.T mutation class in all Cas systems, as shown in *Figure 4a and b* and *Supplementary file 1*.

In clinically significant mutations, the G.C > A.T class predominates the other mutation classes, as shown in *Figure 4b*. There is a clear difference in the abundance of mutational classes among the various Cas systems for clinically relevant SNVs. Thus, Cas systems with G-rich PAMs, such as *Fn*/en*Fn*Cas9 (42.59 and 40.87%), can target G.C > A.T mutation classes, whereas Cas systems with T-rich PAMs, such as *Aa*Cas12b, *Lb*Cas12a, and Cas14a (dsDNA) cannot (37.17, 34.8, and 32.69%). Similarly, *Aa*Cas12b, *Lb*Cas12a, and Cas14a (dsDNA) are enriched for A.T > G.C classes when compared to *Fn*/en*Fn*Cas9, as shown in *Figure 4b*. Since Cas14a (ssDNA) can target nearly all SNVs available both at

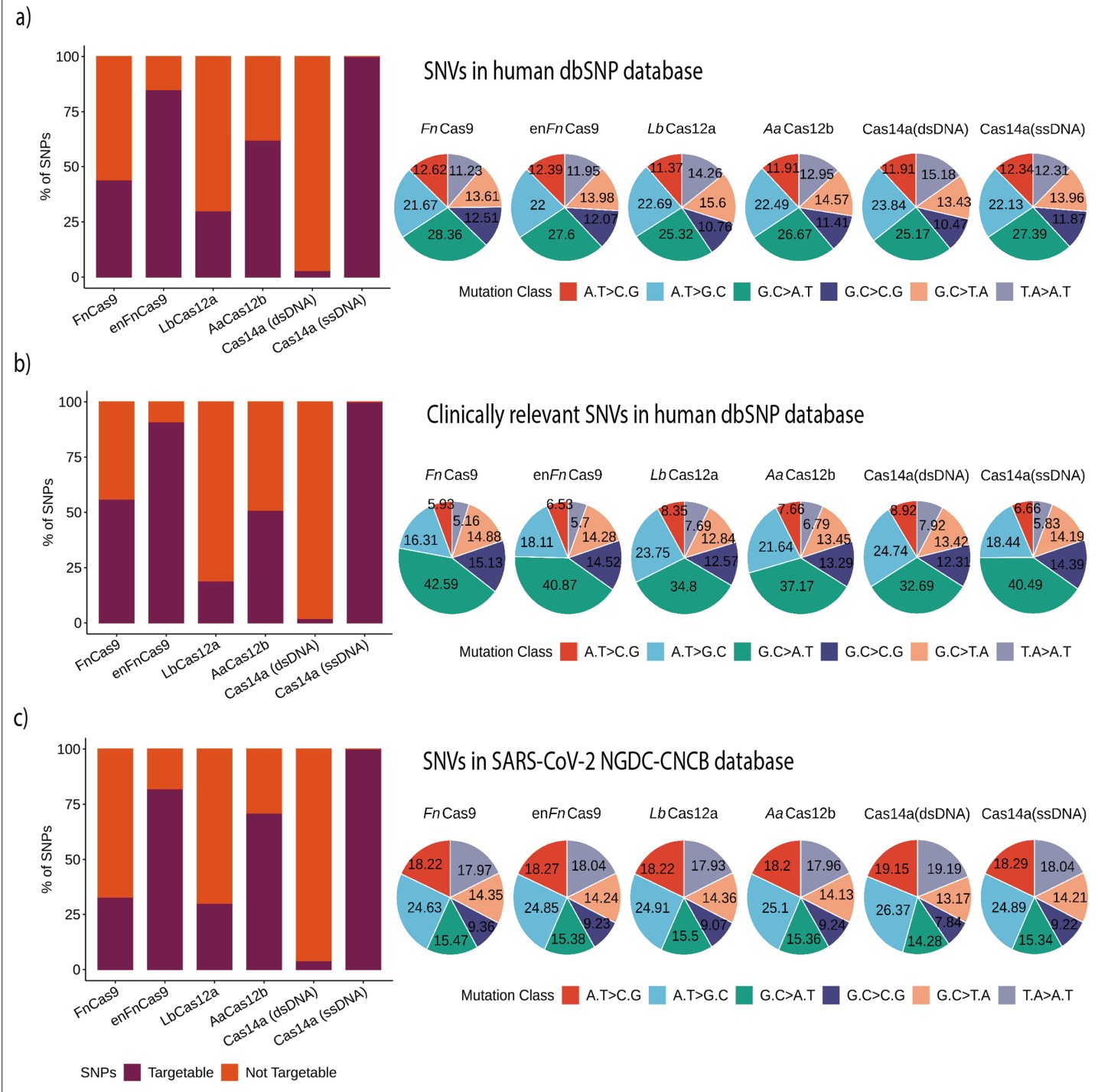

**Figure 4.** Various Cas systems targeting SNPs/single nucleotide variants (SNVs) in SNP Database (dbSNP) and SARS-CoV-2 genomes. (**a**) Shows percent SNP targets for different Cas-systems across the dbSNP, as well as the base distribution of targeted SNPs by individual Cas-system. (**b**) The percentage of targeted SNPs that have clinical significance or disease relevance in humans, with a percentage base distribution at each SNP position targeted by each Cas-system. (**c**) The percentage of targeted SNPs in SARS-CoV-2 genomes reported in the GISAID Database, along with the percentage base distribution at each SNP position targeted by each Cas-system. In all bar plots, red depicts the percentage of non-targeting SNVs, while violet indicates the percentage of SNVs that can be targeted.

The online version of this article includes the following figure supplement(s) for figure 4:

**Figure supplement 1.** Upset plots of the intersection of the targetable variation of various Cas systems.

dbSNP and ClinVar, the mutation classes displayed for Cas14a (ssDNA) can be used as a reference for the other Cas systems in each instance.

Given the ongoing COVID-19 pandemic and the emergence of rapidly mutating SARS-CoV-2 variants, CriSNPr also includes targets for SARS-CoV-2 SNV detection. The CNCB-NGDC SARS-CoV-2 variation database (based on GISAID genome sequences) was used as a reference to create the CriSNPr database. Since the SARS-CoV-2 genome is AT-rich (62.05%) compared to the human genome, this is reflected by decreased targetability for G-rich *Fn*/en*Fn*Cas9 and slightly higher targetability (30 and 70%, respectively) for *Lb*Cas12a and *Aa*Cas12b, *Figure 4c* and *Supplementary file 2*.

## CriSNPr readout can be experimentally validated in a short period of time

We then validated the CriSNPr outputs using real diagnostic assays on two substrates that differed by one mismatch (a mutation corresponding to the E484K signature found in multiple SARS-CoV-2 VOCs). To accomplish this, we purified three of the DNA-targeting effectors, *Fn*Cas9, Cas14a, and *Lb*Cas12a and performed diagnostic assays in accordance with previously published protocols (*Kumar et al., 2021*; *Harrington et al., 2018*; *Teng et al., 2019*). The crRNA sequences obtained from CriSNPr were used to distinguish between WT and mutant sequences via differences in fluorescence intensity generated by trans-cleavage activities of Cas14a and *Lb*Cas12a and on a paper strip (based on affinity-based discrimination) via *Fn*Cas9.

Remarkably, CriSNPr-designed gRNAs were successful in distinguishing between WT and mutant substrates based on the fluorescence intensity of reporter cleavage for both Cas14a and *Lb*Cas12a, as shown in *Figure 5* and *Figure 5—figure supplement 1*. Similar discrimination between WT and E484K mutant sequences was demonstrated by *Fn*Cas9 when used with CriSNPr-derived crRNA sequences for lateral flow assays visualized and quantified using the smartphone app TOPSE (True Outcome Predicted via Strip Evaluation; *Figure 5*, *Figure 5—figure supplement 1*).

For targeting SNPs in the human genome, we considered two common SNPs in the Indian population, rs2073874 (ADAMTSL2, C>T) and rs138739292 (AKAP9, G>A), which are linked to Geleophysic dysplasia 1 and Romano-Ward syndrome, respectively. Through CriSNPr, crRNA and primers were designed for the detection of rs2073874 and rs138739292 by *Fn*Cas9, Cas14a, and *Aa*Cas12b. When selecting the target crRNA, different mismatch positions for individual Cas systems were considered. *Fn*Cas9 could distinguish WT and mutant DNA sequences for both SNPs (*Figure 6a*). As shown in *Figure 6b*, there was a ~20-fold difference in band signal intensity between WT and rs2073874 when a 2 and 6 mismatched crRNA was used. This was ~3.5-fold when mismatches at 16 and 19 positions were considered. Similarly, there was a ~sixfold difference between WT and the rs138739292 mutation (*Figure 6b*). These results supported the previously reported efficacy of 2 and 6 mismatch containing crRNAs for FnCas9-based SNV detection (*Azhar et al., 2021*; *Kumar et al., 2021*). Cas14a, in contrast to FnCas9, functions by collateral cleavage of the fluorescent reporter upon encountering activator ssDNA sequences. It was also able to significantly discriminate between WT and mutant sequences with a mismatched 12th position crRNA for both rs2073874 and rs138739292, with fold differences of ~6.8 and ~4, respectively (*Figure 6c*). Surprisingly, although *Aa*Cas12b has been reported to have the most possible combinations of mismatch-sensitive nucleotide positions, the fold differences of ~1.3- and ~2.3-folds for WT and rs2073874 with previously prioritized mismatch combinations of 1 and 5 and 5 and 11, respectively. Among the Cas proteins that were studied, this difference was the lowest (*Figure 6d*).

Altogether, the in vitro validation experiments with modified crRNA designs generated by CriSNPr demonstrated that the pipeline can design gRNAs targeting SNVs of interest in a reasonable amount of time, as shown in *Figures 5 and 6*. Even though it is possible that some SNVs will require more optimization than others based on the difficulty of amplification and the propensity of individual Cas proteins to discriminate based on design parameters, the elimination of manual design of gRNAs with synthetic mismatches and their off-target information will allow the user to concentrate more on refining assay components. This is especially important during a pandemic or community outbreak of pathogenic variants of the disease.

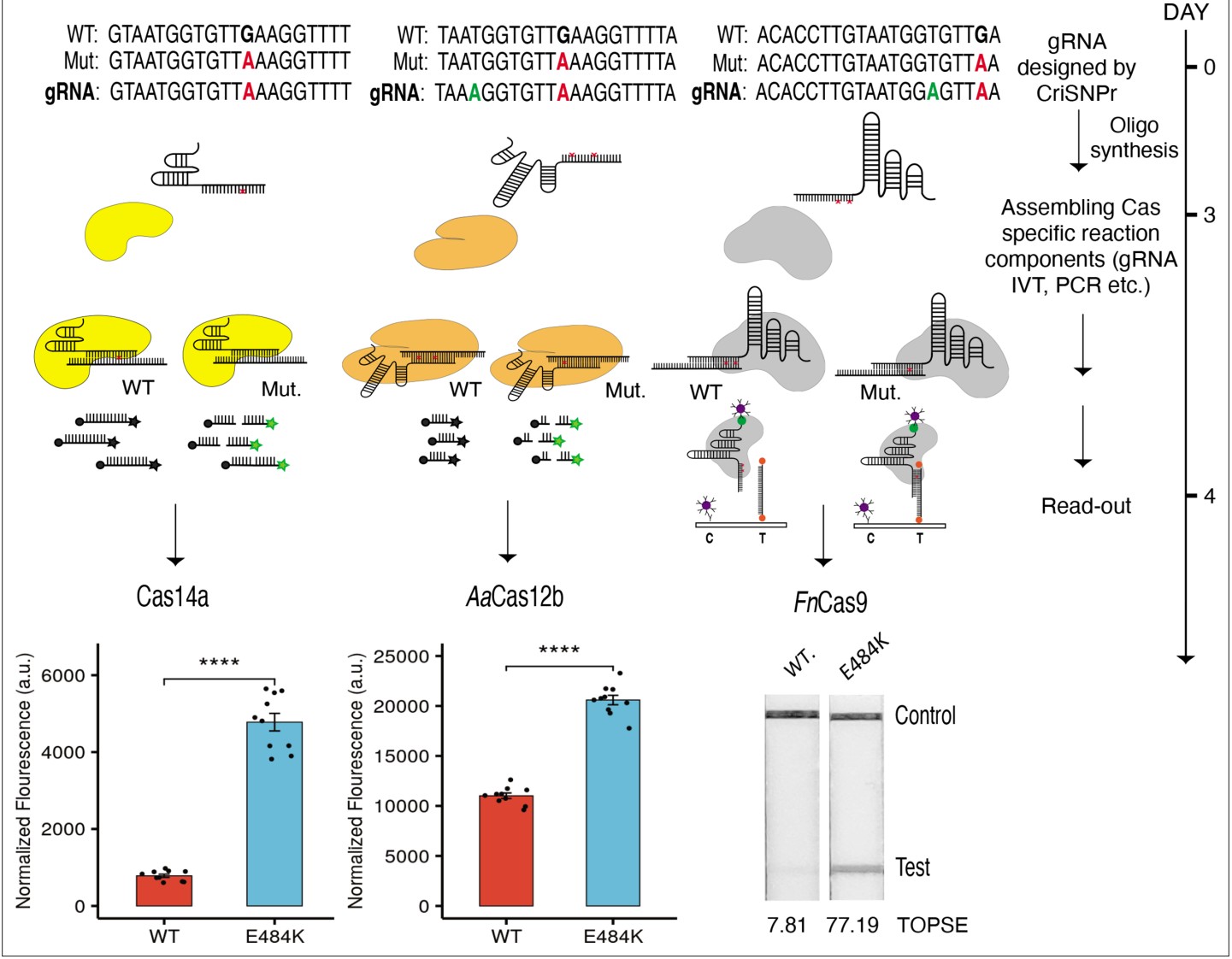

**Figure 5.** CRISPR-based SNP recognition (CriSNPr) designed guide RNAs (gRNAs) can discriminate SARS-CoV-2 single nucleotide variant (SNV) with multiple Cas proteins. crRNA sequences designed by CriSNPr for SARS-CoV-2 E484K variant detection by Cas14a, *Aa*Cas12b, and *Fn*Cas9 can successfully discriminate between wild-type and mutant sequences. A possible implementation schedule for the assays is depicted to the right. SEM, student paired T-test p values **** 0.0001 (dots represent values from independent measurements, n=10).

The online version of this article includes the following source data and figure supplement(s) for figure 5:

**Source data 1.** The red rectangle denotes the approximate area cropped from the LFA strips for generating *Figure 5*.

**Figure supplement 1.** Guide RNA (gRNA) sequences designed by CRISPR-based SNP recognition (CriSNPr) for the detection of the SARS-CoV-2 E484K mutation using Cas14a, *Aa*Cas12b, and *Fn*Cas9, respectively, as shown in *Figure 4* (denoted with a red dotted box).

**Figure supplement 2.** PAGE gel showing purified *Fn*Cas9 (~190kDa), *Aa*Cas12b (~130kDa), and Cas14a (~61kDa) proteins.

**Figure supplement 2—source data 1.** The red rectangle denotes the approximate area cropped from the PAGE gel for generating *Figure 5—figure supplement 2*.

**Figure supplement 2—source data 2.** Original uncropped PAGE gel of purified FnCas9 (190 kDa), AaCas12b (130 kDa), and Cas14a (61 kDa) proteins.

## Discussion

In this manuscript, we present a single web server that provides the user with pre-designed gRNAs and flanking sequences, allowing them to easily construct a CRISPRDx pipeline. In recent years, the field of CRISPR diagnostics has exploded, with multiple Cas systems demonstrating tremendous promise in reading and detecting nucleotide modifications in a substrate (*Li et al., 2018*; *Myhrvold*

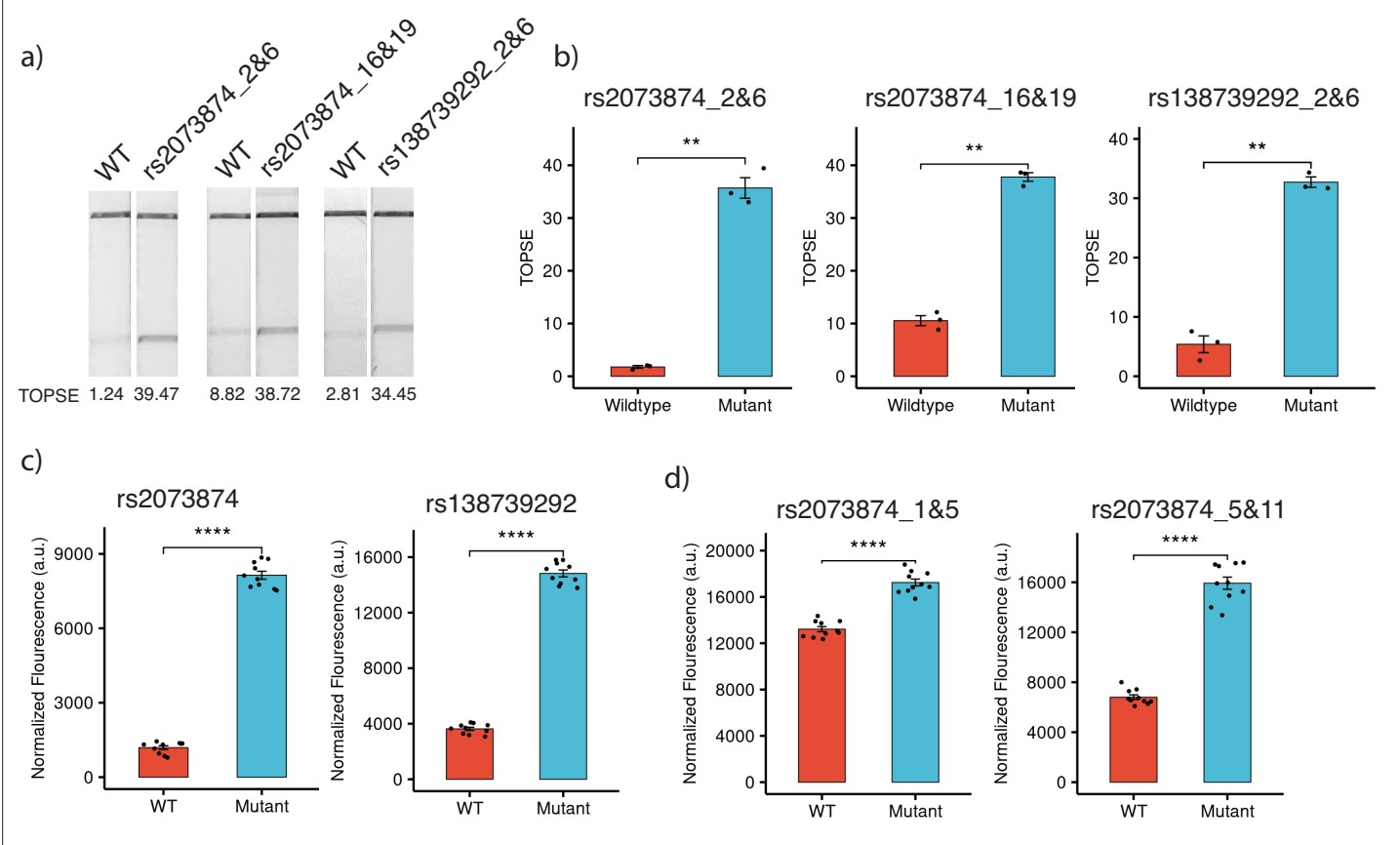

**Figure 6.** Detection of clinically important human SNPs by CRISPR-based SNP recognition (CriSNPr)-designed guide RNAs (gRNAs) for different Cas proteins. (**a**) *Fn*Cas9-based detection of WT (wild-type) and mutant sequences containing rs2073874 and rs138739292 using CriSNPr-designed 2 and 6 and 16 and 19 and 2 and 6 position modified crRNAs, respectively. (**b**) Quantified TOPSE (True Outcome Predicted via Strip Evaluation) intensity values for rs2073874 and rs138739292 detection by *Fn*Cas9, SEM, student paired T-test p values ** 0.01 (dots represent independent measurements, n=3). (**c**) Detection of WT as well as rs2073874 and rs138739292 containing ssDNA sequences using CriSNPr designed gRNAs for Cas14a (dots represent independent measurements, n=10). (**d**) CriSNPr generated 1 and 5 and 5 and 11 modified crRNAs for use with *Aa*Cas12b to distinguish between WT and rs2073874 ssDNA sequences. **** 0.0001 SEM, student paired T-test p values (dots represent independent measurements, n=10).

The online version of this article includes the following source data for figure 6:

**Source data 1.** The red rectangle denotes the approximate area cropped from the LFA strips for generating *Figure 6a*.

*et al., 2018*; *Teng et al., 2019*; *Azhar et al., 2021*; *Harrington et al., 2018*; *Acharya et al., 2019*; *Kumar et al., 2021*; *Acharya et al., 2021*).

As this opens up applications in a variety of biotechnology and clinical diagnostic regimens, particularly for early, on-site detection, the need for streamlined design parameters for assay design is critical. CriSNPr is one of the first web servers aimed at reducing the time and effort required to design CRISPR/Cas-based SNV detection assays. When combined with different readout modalities tailored for multiple CRISPR effectors, it can enable the design of CRISPR diagnostics for the rapid detection of monogenic and infectious diseases in different Cas systems.

While this manuscript was being written, a few web servers that design gRNAs specifically for targeting SNVs across the genome, particularly in an allele-specific manner, were reported in the literature (*Rabinowitz et al., 2020*; *Chen et al., 2020a*; *Zhao et al., 2020*; *Keough et al., 2019*). These present an important advancement toward gRNA design for precision medicine in general. However, the CriSNPr platform is tailored for generating gRNAs specifically for diagnostics, taking into account the design parameters for each diagnostic CRISPR protein. Thus, it caters to applications that are not covered by general gRNA design databases and toolsets (*Supplementary file 3*).

In the current version of CriSNPr, six of the widely used CRISPR diagnostic platforms have been added because there is enough research to back up their design guidelines. CRISPR diagnostics is a rapidly evolving field; recently, several other Cas proteins have been reported for identifying

pathogenic DNA or RNA and their variants (*Fasching et al., 2022*; *Nan et al., 2022*). As more literature supporting these diagnostic pipelines became available, we would incorporate it into the server. The inclusion of multiple CRISPR pipelines on an integrated server serves two purposes. To begin, an SNV of interest may be detected only by certain Cas proteins and not by others, as shown in *Supplementary file 1*. Second, when multiple diagnostic options are available, users can choose their preferred CRISPR platform based on the availability of reagents and methodologies. The latter is especially useful because the sensitivity and scope of readout modes can differ depending on the diagnostic query: pathogenic polynucleotide or single nucleobase variant (monoallelic or biallelic; *Arizti-Sanz et al., 2020*; *Joung et al., 2020*). While lateral flow readouts have been demonstrated for detecting full-length sequences and variants, fluorescence-based readouts are better suited for detecting disease mutation carriers (monoallelic SNVs; *Li et al., 2018*; *Myhrvold et al., 2018*; *Teng et al., 2019*; *Azhar et al., 2021*; *Harrington et al., 2018*; *Kumar et al., 2021*; *Kellner et al., 2019*; *Gootenberg et al., 2018*). CriSNPr recommendations for the given SNV of interest can help with such decisions. Importantly, a wide range of design options allows users to test and standardize the best pipeline for their chosen SNV. This is especially true because the various CRISPR systems with PAM requirements prefer AT/GC-rich PAM sequences. As the number of such sequences differs between human and other pathogenic genomes, so does the ability of CRISPR proteins to distinguish between SNVs based on the target species.

There are a variety of ways in which CriSNPr could be improved immediately. For instance, the current framework of CriSNPr cannot perform sequence batch processing. This limitation is due to the technical parameters of the server's hosting system, which will be updated in the near future to accommodate batch processing. Furthermore, CriSNPr currently does not include variants that are not single nucleotide changes. Although SNVs are the most sensitive and critical diagnostic challenge, later versions of CriSNPr will include gRNA design parameters for polynucleotide changes in the target.

In our study, the importance of certain mismatch-sensitive nucleotide positions for the detection of SNV has been emphasized, making their selection crucial when determining a mismatch combination. When scoring sgRNAs, modified crRNAs that have been shown to be experimentally efficient with their respective Cas effectors are given priority. This suggests that in several cases, the design strategy used to assign mismatches in gRNAs will be greatly improved by further experimental validation of gRNA mismatches by multiple Cas species. This is especially true for several of the Cas effectors considered in this server, such as *Fn*Cas9, Cas14, and Cas13, where systematic dissection of every combination of nucleobase mismatch on gRNA sensitivity has not yet been reported (*Myhrvold et al., 2018*; *Azhar et al., 2021*; *Harrington et al., 2018*; *Kumar et al., 2021*; *Kellner et al., 2019*; *Acharya et al., 2021*). As more such data becomes available, we will be able to improve our scoring of gRNAs returned as output, allowing users to spend less time validating more efficient pipelines.

CriSNPr currently hosts genomic targets for humans and SARS-CoV-2 lineages. This is a limitation given that a large number of pathogenic variants exist in the microbial community, and a number of these have significant clinical manifestations, such as drug or antibiotic resistance. Although the web server has a de novo design feature, we hope to expand the scope of the tool to include as many sequences that can be retrieved from public repositories as possible. Perhaps the most critical evaluation of CriSNPr will be possible once more such web servers are made available for comparison. Until that happens, CriSNPr, in its current form, should help implement CRISPR diagnostics in a wide range of clinical and academic settings and reduce the time and effort needed for design and validation for every nucleotide variant.

## Materials and methods
### Oligos
A list of all oligomers (Merck) used in the study can be found in *Supplementary file 4*, along with a figure-wise mention of their use.

## Generation of the CriSNPr database

### For the human SNPs/SNVs with pathological relevance

To build the CriSNPr database for variant detection through various Cas systems for population-specific SNPs and pathological mutations in humans, the latest dbSNP build 155 (GCF 000001405.39) was downloaded from the NCBI FTP site (*Sherry et al., 2001*). Since the focus was on SNPs/SNVs, all other variation types like insertion, deletion, duplication, and translocation were filtered out, leaving only variants with a common SNV tag and valid ClinVar ID for further analysis (*Landrum et al., 2020*; *Landrum et al., 2018*). vcflib (v1.0.0) was used to convert vcf into a tabular format for smooth analysis while retaining trivial information such as chromosome coordinates, reference and alternate alleles, reference SNP (rsID), gene, disease, and population level frequency, and so on *Garrison et al., 2021*. Besides that, multi-allelic SNPs were split into multiple entries for easier processing and labeled as 'not available' if missing or 'not provided' if the information was unavailable. All previous data filtrations and curations were carried out using the Pandas Python library (v1.3.5; *McKinney, 2010*). Finally, to design the gRNAs for detection, filtered SNVs/SNPs positions were mapped to the human reference genome (Gencode GRCh38.p13) to cross-check that the reference base lies within the PAM proximity, followed by the incorporation of a synthetic mismatch at positions based on the use of different Cas systems (*Figure 1*; *Harrow et al., 2012*; *Frankish et al., 2019*; *Frankish et al., 2021*). Furthermore, genome coordinates for the target SNV were used to obtain flanking sequences for the design of target amplification primers. Since *Fn*Cas9 does not have any collateral activity, primers were made so that in vitro cleaved products after an enzyme treatment can be optimally resolved, keeping the lengths of the cleaved products in a ratio of 1:3 to 2:3.

All of this was accomplished by utilizing the BioPython SeqIO library (v1.76) for genome parsing, followed by PAM allocation by using the regex library (v2021.11.10) (*Cock et al., 2009*). Nucleotide base incorporation was done by creating a custom function. The entire CriSNPr database was created using Python's SQLite3 library (v3.31.1; *Van Rossum and Drake, 2009*; *SQLite, 2020*). Real-time SNP frequency data is plotted as a bar graph with the matplotlib library (v3.1.3; *Hunter, 2007*). A custom Python function using the Primer3 library (v0.6.1) is developed to generate amplicon primers (*Untergasser et al., 2012*).

### For the SARS-CoV-2 variants

CNCB-NGDC (China National Center for Bioinformation-The National Genomics Data Center) nucleotide level variant annotation data was downloaded, which provided the most recent variant information based on an analysis of the GISAID genome sequences dataset (*Xue et al., 2021*; *Shu and McCauley, 2017*). After downloading the variant annotation data table, gene and transcript mutations were analyzed, and variations in intergenic and untranslated regions were removed. Non-SNV mutations were then filtered out of the data. Information like chromosome, position, reference allele, alternate allele, amino acid change, and variation frequency was included for further analysis. In real time, the number of virus sequences with an SNP is plotted as a bar plot using the matplotlib library (*Hunter, 2007*). The standard CoV-2 variation nomenclature requires a gene name followed by an underscore, a reference amino acid, its position on the protein, and an alternative amino acid. So, a column was added for the variation identity that could be used as a query by the user. The target gRNAs and primers for SARS-CoV-2 variants were designed in the same manner as previously described for the human dbSNPs dataset.

## Generation of seq-CriSNPr

Seq-CriSNPr is a real-time sequence variant detection tool that employs crRNA and primer design. The user must enter the sequence, variant position, and identity. Initially, the identity of the variant base was distinguished in relation to the reference base at the user-supplied position. If the variant base identity is the same as the reference base, the back-end server returns an error message. Additionally, Flask-WTForms checks an input sequence for the invalid base ('non-ACGT' characters) and length (20–30 nucleotides; v2.2.1). The sequence is then mapped to the user-selected organism reference genome using the BWA aligner with no mismatch to obtain the variant position on the genome (*Li and Durbin, 2009*; *Li and Durbin, 2010*). If the sequence is not found in the genome as defined by the user, it is returned with a warning. Furthermore, the variant's genome position is queried against the existing CriSNPr database; if not found, PAM sequences are searched nearby as defined

in *Figure 1*. Upon PAM localization, the variant base is inserted into the sequence, followed by a random mismatch at the appropriate position, as shown in *Figure 1*. BEDTools (v2.29.2) is used to extract the gene annotation of the crRNA location (*Quinlan and Hall, 2010*). Following that, primers were designed using the Primer3 Python library (v0.6.1), as previously described (*Untergasser et al., 2012*).

### Off-target prediction and scoring of modified gRNAs

CriSNPr simultaneously creates an off-target link for each crRNA design. After clicking the link, based on the organism and Cas-system, a request is sent to the back end, and a file is generated in Cas-OFFinder (v2.4) format (*Bae et al., 2014*). Following that, a customized function is written to generate a file format compatible with the Cas-OFFinder and predict off-targets with up to four mismatches by the Cas-OFFinder stand-alone version in CPU mode. These off-targets are sorted ascendingly based on the number of mismatches to the crRNA sequence. The number of off-targets is then used to score the crRNA with the fewest or no off-targets.

### Development of the CriSNPr web server

The web server was designed and built using Flask (v1.1.1), Jquery (v3.5.1), and Bootstrap (v5.0.2; *Grinberg, 2014*). All codes were written in Python (v3.7.6) and were maintained using the Conda environment (v4.11.0; *Van Rossum and Drake, 2009*; *Anaconda, 2016*). CriSNPr's standalone (or offline) version was developed using Python and Conda. The full source code for the pipeline is available at https://github.com/asgarhussain/CriSNPr, (*Ansari, 2023*).

### Percentage targetability of SNVs in humans and SARS-CoV-2 by individual Cas systems

The CriSNPr database contains crRNA for a variety of genes that can be used to detect SNPs. Data on the targetability of different Cas systems were computed with the help of the CriSNPr database in the Pandas library and visualized in R (v4.0.5) with the help of the ggpubr (v0.4.0) package (*R Development Core Team, 2017*). Different mutation classes of various Cas systems were also analyzed in Pandas (v1.3.5) and plotted with the ggpubr R package, as shown in *Figure 4* (*Supplementary files 1 and 2*; *McKinney, 2010*; *R Development Core Team, 2017*). The intersections of target variations from various Cas systems were calculated using Upset modules from the Intervene offline version (v0.6.5; *Khan and Mathelier, 2017*; *Lex et al., 2014*).

### Protein purification

A plasmid containing *Fn*Cas9 or d*Fn*Cas9 (dead or catalytically inactive; *Acharya et al., 2019*) was transformed and expressed in *Escherichia coli* Rosetta 2 (DE3) (Novagen). Transformed Rosetta 2 (DE3) cells were cultured at 37°C in an LB medium containing 50 mg/ml kanamycin until the OD600 reached 0.6. After inducing protein expression with 0.5 mM isopropyl b-D-thiogalactopyranoside (IPTG), the culture was grown overnight at 18°C. Centrifuged *E. coli* cells were lysed by sonication in a buffer (20 mM HEPES, pH 7.5, 500 mM NaCl, 5% glycerol, and 100 mg/ml lysozyme) supplemented with 1× PIC (Roche). The supernatant obtained after centrifugation was then passed through Ni-NTA beads (Roche) and eluted with a buffer (20 mM HEPES, pH 7.5, 300 mM imidazole, and 500 mM NaCl). The eluted fractions were concentrated, and the protein was further purified with size-exclusion chromatography on a HiLoad Superdex 200 16/60 column (GE Healthcare). Finally, the protein was quantified using the Pierce BCA protein assay kit (Thermo Fisher Scientific) and stored in a buffer solution (20 mM HEPES pH 7.5, 150 mM KCl, 10% glycerol, and 1 mM DTT) at –80°C until use in a reaction (*Figure 5—figure supplement 2*).

Proteins *Aa*Cas12b and Cas14a were purified using previously published protocols, with any necessary modifications (*Teng et al., 2019*; *Harrington et al., 2018*). *Aa*Cas12b (Addgene no. 113433) and Cas14a (Addgene no. 112500) plasmids were transformed and expressed in *E. coli* BL21 (DE3; Novagen). Transformed BL21 (DE3) cells were cultured at 37°C in terrific broth medium with appropriate antibiotics and induced with 0.5 mM IPTG when the OD600 reached 0.6. Overnight cultures at 18°C were harvested, and cells were lysed with sonication in a buffer (50 mM Tris-HCl, pH 7.5, 5 mM Imidazole, and 500 mM NaCl) supplemented with 1× PIC (Roche). Following centrifugation, the supernatant was passed through Ni-NTA beads (Roche) and washed with a wash buffer (50 mM

Tris-HCl, pH 7.5, 20 mM imidazole, and 500 mM NaCl). The proteins were then eluted with a buffer (50 mM Tris-HCl, pH 7.5, 300 mM imidazole, and 500 mM NaCl), and MBP and his tags were removed by overnight TEV (tobacco etch virus) protease incubation at 4°C. Size-exclusion chromatography on a HiLoad Superdex 200 16/60 column (GE Healthcare) was used to purify the filtered proteins, which were then analyzed using SDS-PAGE. Following quantification, the purified *Aa*Cas12b and Cas14a were then stored in a buffer (20 mM Tris-HCl, pH 7.5, 250 mM NaCl, 5% glycerol, and 1 mM DTT) at –80°C until use (*Figure 5—figure supplement 2*).

## SARS-CoV-2 and human SNP/SNV detection through CriSNPr designed gRNAs for AaCas12b, Cas14a, and FnCas9

### FnCas9 - RAY

CriSNPr was employed to generate modified gRNA and primers based on the SNV-containing regions of the SARS-CoV-2 and human genomes, respectively. Reverse-transcribed and 5'-end biotin-labeled amplicons with or without mutation were used as target sequences. The chimeric gRNA was made by equimolar mixing crRNA and synthetic 3'-FAM-labeled tracrRNA in a buffer (100 mM NaCl, 50 mM Tris-HCl pH 8, and 1 mM $MgCl_2$) and heating at 95°C for 2–5 min, followed by slow cooling for 15–20 min at room temperature. Following that, equimolar gRNA:dFnCas9 RNP complexes were prepared in a buffer (20 mM HEPES, pH 7.5, 150 mM KCl, 1 mM DTT, 10% glycerol, and 10 mM $MgCl_2$) and incubated for 10 min at room temperature. When active *Fn*Cas9 was used, $MgCl_2$ was removed from the buffer, rendering it catalytically inactive. Finally, the target 5' end biotin-labeled amplicons were treated with the RNP complexes for 10 min at 37°C, and the readout was obtained by adding 80 µl of dipstick buffer and a Milenia Hybrid 1 lateral flow assay strip should be detected for 2–5 min at room temperature before visual or smartphone app-based (TOPSE) quantification (*Azhar et al., 2021*; *Kumar et al., 2021*).

### AaCas12b - CDetection

AaCas12b-based FQ detection was carried out using active RNP, which was prepared by equimolar mixing and incubating *Aa*Cas12b and gRNA in a buffer (40 mM Tris-HCl, pH 7.5, 60 mM NaCl, and 6 mM $MgCl_2$) for 10 min at room temperature. Next, ssDNA target (60 nt WT/mutant) sequences mixed with background human genomic DNA and custom synthesized homopolymer (poly T) 5 nt ssDNA FQ reporter molecules (GenScript) were added to the reaction in a Corning 96-well flat-bottom black clear bottom microplate. Reactions were left at 37°C for the given times (up to 180 min) in a fluorescence plate reader (Tecan). With fluorescence intensity measured every 10 min ($\lambda_{ex}$: 490 nm; $\lambda_{em}$: 520 nm, transmission gain: optimal), the resulting data after background subtraction using intensity values recorded in the absence of ssDNA target sequences were plotted using an R script (*Teng et al., 2019*).

### Cas14a - DETECTR

Active Cas14a RNP complexes used for FQ detection were made by equimolar mixing Cas14a and sgRNA in a buffer (40 mM Tris-HCl, pH 7.5, 60 mM NaCl, and 6 mM $MgCl_2$) and incubating for 10 min at room temperature. Furthermore, fluorescence readouts were created by combining ssDNA (60 nt WT/mutant) target sequences with background human genomic DNA and 200 nM 12 nt poly T ssDNA FQ reporter molecules (GenScript) in a Corning 96-well flat-bottom black clear bottom microplate. Reactions were performed at 37°C for the points in time described (up to 180 min) in a fluorescence plate reader instrument (Tecan). The intensity of fluorescence was measured every 10 min ($\lambda_{ex}$: 490 nm; $\lambda_{em}$: 520 nm; transmission gain: optimal). Finally, the data obtained after background subtraction (intensity in the absence of ssDNA target sequences) was plotted using an R script (*Harrington et al., 2018*).

## Acknowledgements

We thank all members of the Chakraborty and Maiti labs for helpful discussions and valuable insights about this work. This study was funded by CSIR-Sickle Cell Mission (HCP0023) and EMBO Young Investigator award, a Lady Tata Young Investigator award (GAP0198) to DC.

## Additional information

### Funding

| Funder | Grant reference number | Author |
| --- | --- | --- |
| EMBO | GAP252 | Debojyoti Chakraborty |
| Lady Tata Memorial Trust | GAP198 | Debojyoti Chakraborty |
| CSIR-Sickle Cell Mission | HCP0023 | Souvik Maiti<br>Debojyoti Chakraborty |

The funders had no role in study design, data collection and interpretation, or the decision to submit the work for publication.

### Author contributions

Asgar H Ansari, Conceptualization, Resources, Software, Formal analysis, Investigation, Methodology, Writing – original draft, Writing – review and editing; Manoj Kumar, Conceptualization, Resources, Data curation, Formal analysis, Validation, Methodology, Writing – original draft, Writing – review and editing; Sajal Sarkar, Resources, Data curation, Formal analysis, Validation, Investigation, Methodology, Writing – review and editing; Souvik Maiti, Conceptualization, Supervision, Funding acquisition, Investigation, Project administration, Writing – review and editing; Debojyoti Chakraborty, Conceptualization, Supervision, Funding acquisition, Investigation, Methodology, Writing – original draft, Project administration, Writing – review and editing

### Author ORCIDs

Asgar H Ansari ⓘ http://orcid.org/0000-0002-1172-9521
Manoj Kumar ⓘ http://orcid.org/0000-0003-0772-1399
Debojyoti Chakraborty ⓘ http://orcid.org/0000-0003-1460-7594

### Decision letter and Author response

Decision letter https://doi.org/10.7554/eLife.77976.sa1
Author response https://doi.org/10.7554/eLife.77976.sa2

## Additional files

### Supplementary files

• Supplementary file 1. Variation statistics from human SNP Database (dbSNP) for various Cas systems.

• Supplementary file 2. Variation statistics from SARS-CoV-2 CNCB-NGDC database for various Cas systems.

• Supplementary file 3. Comparison between different sgRNA designing tools for single nucleotide variants (SNVs) in a sequence.

• Supplementary file 4. List of oligos used in this study.

• Transparent reporting form

### Data availability

The current manuscript is a computational study, so no new data has been generated for this manuscript. Experimental validation results have been presented in figures in the manuscript. The source code and related datasets have been indicated in the manuscript and also uploaded here: http://crisnpr.igib.res.in/. All other validation data have been presented in the main manuscript itself.

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
