## [Editor Report]

The web-based software developed in this study will be of interest to researchers who develop CRISPR-based diagnostic methods. The use of CRISPR-Cas to rapidly identify specific mutations in both cancer and infection is an evolving field with good potential to play a role in future research and diagnostics. This software will facilitate the implementation of such technologies and is therefore useful.

---

## [Decision Letter]

**Decision letter after peer review:**

Thank you for submitting your article "CriSNPr: a single interface for the curated and de-novo design of gRNAs for CRISPR diagnostics using diverse Cas systems" for consideration by *eLife*. Your article has been reviewed by 3 peer reviewers, and the evaluation has been overseen by a Reviewing Editor and Mone Zaidi as the Senior Editor. The following individuals involved in review of your submission have agreed to reveal their identity: Shruthi Sridhar Vembar (Reviewer #1).

Essential revisions:

While the authors have provided a small set of data that validated the design of the software, more validation experiments will be needed to demonstrate how well the guide RNAs work, or which designs are predicted to work better than others. These additional data should further support the usability of the software.

1) Figure 4 gives one example with the detection of a short WT and mutated oligo from SARS-CoV-2 as the template. Similar experiments may be done with several SNVs in real cells, to also include the complexity of genomic DNA. For comparison, it would be elegant to include a gRNA also recognizing the WT sequence in some of the experiments, including in Figure 4.

2) The authors only test their system for a single mutation (E484K) with three CRISPR effector proteins (FnCas9 and LbCas12a). More extensive validation (multiple designs per mutation, many mutations examined) is crucial for sequence design methods like CriSNPr.

3) How well do the pre-designed gRNAs work for detecting human SNPs? The authors should compare their method with existing assays to detect human SNPs.

*Reviewer #1 (Recommendations for the authors):*

I congratulate the authors on a great piece of work. Here are some suggestions which could improve the manuscript:

1. The Yes/No options in the flowchart of Figure 3a were confusing. For example, for the Input SNP ID box, the first check is if it is valid or not. The flowchart uses 'Not Valid' instead of 'Valid', and hence there is a double negative which becomes a positive, and so on. My suggestion is to change 'Not Valid' to 'Valid or not': if the answer to this is 'Valid', the Python Flask is invoked; if it is 'Not valid', the program outputs an error message and exits. The same is true for the 'Not found' option. I am sorry if this point doesn't make sense, but if I could have included a figure here, it would have been easier to explain.

2. To improve the flow of the paper, figures 4 and 5 could be switched. Basically, the paper could end with the experimental detection of the SARS-CoV-2 variant, S gene containing E484K mutation, using CriSNPr-designed CRISPR RNAs and PCR primers.

3. Since page and line numbers were not included in the submitted manuscript, it is difficult for me to point out the sentences which could be reworded for clarity. There were also a few typos and phrase/word repetitions that should be addressed.

One last suggestion is to develop a stand-alone version of CriSNPr which researchers can download and work with locally.

For the online version, please include a form in the 'Contact us' section where researchers could write to the authors requesting their organism of choice to be included in CriSNPr.

*Reviewer #2 (Recommendations for the authors):*

Here follows some suggestions:

It makes the review process easier if the manuscript has line numbers. I suggest adding this in the future to any manuscripts.

I realize that this could be difficult, but is there anyway the user can be helped to prioritize which gRNAs to use? Is there any scoring system that could be used or developed? It would be useful to at least discuss this a bit in the manuscript.

As briefly stated above, the manuscript would benefit from more validation experiments, in a systematic way showing to what level the suggested guide RNAs work. Figure 4 gives one example with the detection of a short WT and mutated oligo from SARS-CoV-2 as the template. I would suggest doing similar experiments also with several SNVs in real cells, to also include the complexity of genomic DNA. For comparison, it would be elegant to include a gRNA also recognizing the WT sequence in some of the experiments, including in Figure 4.

The manuscript would benefit from more detailed information about the basis for the guide RNA design for the different Cas proteins, including highlighting references better in the methods part as well as elsewhere in the manuscript (e.g. on p. 5 "… and the required crRNA and primer design parameters based on gRNA design principles available in the literature for each Cas protein" would benefit from adding references).

I lack references on other places too:

On p. 20 "performed the diagnostic assays according to previously published protocols".

On p. 23 "In clinically relevant variations, G.C>A.T class is dominating the other mutation classes".

On several places in the manuscript, its stated that specific updates will be made to the software (e.g. page 26). On the one hand I appreciate the dedication, but it also begs the question if they really think the software is ready to be published already.

Figure 3 and Sup. figure 2, are of fairly low quality. I realize that this is because the figure has been copied from the webpage, but consider if you can make some changes so that the quality is higher.

Consider rearranging the first page of the web page. To me it doesn't make sense that the "Seq-CriSNPr" part is shifted to the right compared to the "human" part. It also looks a bit strange that the distance on the y-axis between the "Seq-CriSNPr" and "Human", is much longer than the distance between "CriSNPr" and "Human". I think the web page will look more professional if this is fixed.

*Reviewer #3 (Recommendations for the authors):*

1. The first three figures are essentially schematics. There isn't enough data in this paper to support the claims made by the authors. The authors should provide additional data showing how they tested their system (both computationally and experimentally).

2. The authors only test their system for a single mutation (E484K) with three CRISPR effector proteins (FnCas9 and LbCas12a). More extensive validation (multiple designs per mutation, many mutations examined) is crucial for sequence design methods like CriSNPr.

3. How well do the pre-designed gRNAs work for detecting human SNPs? The authors should compare their method with existing assays to detect human SNPs.

---

## [Author Response]

Reviewer #1 (Recommendations for the authors):I congratulate the authors on a great piece of work. Here are some suggestions which could improve the manuscript:1. The Yes/No options in the flowchart of Figure 3a were confusing. For example, for the Input SNP ID box, the first check is if it is valid or not. The flowchart uses 'Not Valid' instead of 'Valid', and hence there is a double negative which becomes a positive, and so on. My suggestion is to change 'Not Valid' to 'Valid or not': if the answer to this is 'Valid', the Python Flask is invoked; if it is 'Not valid', the program outputs an error message and exits. The same is true for the 'Not found' option. I am sorry if this point doesn't make sense, but if I could have included a figure here, it would have been easier to explain.

We thank the reviewer for this suggestion, and we understand what the reviewer is trying to convey. We have updated this in the manuscript accordingly by replacing the Valid/Not valid with Yes/No responses.

2. To improve the flow of the paper, figures 4 and 5 could be switched. Basically, the paper could end with the experimental detection of the SARS-CoV-2 variant, S gene containing E484K mutation, using CriSNPr-designed CRISPR RNAs and PCR primers.

We thank the reviewer for this suggestion, and the manuscript is updated accordingly by switching the two figures as suggested.

3. Since page and line numbers were not included in the submitted manuscript, it is difficult for me to point out the sentences which could be reworded for clarity. There were also a few typos and phrase/word repetitions that should be addressed.

We apologize for typographical errors and repetitions. We have now corrected and proof-read them in the current version of the manuscript.

One last suggestion is to develop a stand-alone version of CriSNPr which researchers can download and work with locally.

We thank the reviewer for this suggestion; an online request form is now added in the "Contact Us" section for the user requests to include new genomes. To further improve flexibility, the stand-alone version of the CriSNPr is also updated with an option to include newer genomes based on the user’s choice.

For the online version, please include a form in the 'Contact us' section where researchers could write to the authors requesting their organism of choice to be included in CriSNPr.

We thank the reviewer for this suggestion; an online request form is now added in the "Contact Us" section for the user requests to include new genomes. To further improve flexibility, the stand-alone version of the CriSNPr is also updated with an option to include newer genomes based on the user’s choice.

Reviewer #2 (Recommendations for the authors):Here follows some suggestions:It makes the review process easier if the manuscript has line numbers. I suggest adding this in the future to any manuscripts.I realize that this could be difficult, but is there anyway the user can be helped to prioritize which gRNAs to use? Is there any scoring system that could be used or developed? It would be useful to at least discuss this a bit in the manuscript.

We thank the reviewer for these excellent suggestions, and as the reviewer expected, it was difficult to prioritize crRNAs because different Cas effectors have different mismatch-sensitive nucleotide positions. However, we have now implemented a scoring method that prioritizes crRNAs with the lowest number of off-targets in the target genome, and if off-targets are high for a crRNA with any combinatorial positions, the next set of positions is automatically preferred.

As briefly stated above, the manuscript would benefit from more validation experiments, in a systematic way showing to what level the suggested guide RNAs work. Figure 4 gives one example with the detection of a short WT and mutated oligo from SARS-CoV-2 as the template. I would suggest doing similar experiments also with several SNVs in real cells, to also include the complexity of genomic DNA. For comparison, it would be elegant to include a gRNA also recognizing the WT sequence in some of the experiments, including in Figure 4.

We thank the reviewer for these suggestions. We have now added more pathogenic SNP targets, made crRNAs for multiple Cas proteins using CriSNPr, tested them, and included them in Figure 6 of the new version of the paper. The target SNPs (rs2073874 and rs138739292) were selected based on their high prevalence in the Indian population and their targetability with different Cas systems.

The reviewer has also mentioned doing similar experiments inside cells to negate the effects of genomic DNA complexity, if any. We would like to humbly point out that all CRISPR diagnostics described so far are done exclusively in vitro and in a non-invasive manner where the complexity of genomic DNA doesn’t come into the picture due to prior amplification steps.

The reviewer has suggested including a gRNA recognizing the WT sequence. We would like to point out that there are multiple existing software pipelines (CRISPOR, Chop-Chop etc.) that design gRNAs against a given WT sequence. The objective of CriSNPr is not to design de-novo gRNA sequences against a target but to design gRNAs that can distinguish a WT sequence from its mutant counterpart.

The manuscript would benefit from more detailed information about the basis for the guide RNA design for the different Cas proteins, including highlighting references better in the methods part as well as elsewhere in the manuscript (e.g. on p. 5 "… and the required crRNA and primer design parameters based on gRNA design principles available in the literature for each Cas protein" would benefit from adding references).

We thank the reviewer for pointing this out and apologize for this inadvertent error. We have corrected this in the updated manuscript and added all the necessary references.

I lack references on other places too:On p. 20 "performed the diagnostic assays according to previously published protocols".On p. 23 "In clinically relevant variations, G.C>A.T class is dominating the other mutation classes".

We apologize once again for these errors; references have now been updated in the current version of the manuscript. On p. 23, the statement is supported by the analysis done in Figure 4b showing the percentage of targeted SNPs having clinical significance or disease relevance in humans, again with percentage base distribution at each SNP position targeted by individual Cas systems.

On several places in the manuscript, its stated that specific updates will be made to the software (e.g. page 26). On the one hand I appreciate the dedication, but it also begs the question if they really think the software is ready to be published already.

We thank the reviewer for bringing this to our attention. The current version of CriSNPr is fully functional to design crRNA and primers for *Fn*/en*Fn*Cas9, *Lw*Cas13a, *Lb*Cas12a, *Aa*Cas12b, and Cas14a. As a field, novel CRISPR diagnostic methodologies are continuously evolving, and thus the statement about updates meant that such methodologies may be incorporated into the pipeline as these technologies mature. We apologize if this meaning was not clearly communicated to the reviewer.

That said, the stand-alone version of CriSNPr with the batch-processing mode is included in the updated version of our manuscript. We have also made crRNA databases for humans and SARS-CoV-2 to speed up the rate of data output.

Figure 3 and Sup. figure 2, are of fairly low quality. I realize that this is because the figure has been copied from the webpage, but consider if you can make some changes so that the quality is higher.

We apologize for this inadvertent error and have corrected it in the revised version of the manuscript.

Consider rearranging the first page of the web page. To me it doesn't make sense that the "Seq-CriSNPr" part is shifted to the right compared to the "human" part. It also looks a bit strange that the distance on the y-axis between the "Seq-CriSNPr" and "Human", is much longer than the distance between "CriSNPr" and "Human". I think the web page will look more professional if this is fixed.

We thank the reviewer for pointing this out and have made the suggested changes to make it look more professional.

Reviewer #3 (Recommendations for the authors):1. The first three figures are essentially schematics. There isn't enough data in this paper to support the claims made by the authors. The authors should provide additional data showing how they tested their system (both computationally and experimentally).

The reviewer has mentioned the paucity of data in this paper to support the claims. We agree that the first figure is a schematic to introduce the concept of CRISPR-based detection platforms to readers. But we respectfully disagree with the idea that the second and third figures should just be called schematics. The second figure depicts the entire pipeline that has been optimized to create the CriSNPr database for both humans and SARS-CoV-2 and highlights key steps that are being taken. While the second figure shows the pipeline, the third figure introduces the front end of CriSNPr as a web server and highlights its utilities, which are important to avail crRNA and primer designs for developing SNP-detection assays.

To address the concern about the paucity of validation data, we now include more data on additional SNVs in Figures 5 and 6 to conclusively establish the utility of criSNPr for designing CRISPR diagnostic assays for pathogenic and non-pathogenic SNVs.

2. The authors only test their system for a single mutation (E484K) with three CRISPR effector proteins (FnCas9 and LbCas12a). More extensive validation (multiple designs per mutation, many mutations examined) is crucial for sequence design methods like CriSNPr.

We thank the reviewer for pointing this out; we have now added more target human SNPs and validated them with all three Cas systems (*Fn*Cas9, Cas14a, and *Aa*Cas12b) in Figure 6.

3. How well do the pre-designed gRNAs work for detecting human SNPs? The authors should compare their method with existing assays to detect human SNPs.

We would like to humbly point out that the pre-designed sgRNAs validated in this manuscript are indeed for human SNPs. However, CriSNPr is not a new platform for CRISPRDx, as the reviewer has probably pointed out. The CriSNPr algorithm uses crRNA design parameters reported for different Cas systems for SNP detection. Its utility is in providing a single platform for designing these for different SNVs as well as testing multiple CRISPR systems for the same SNV. In this way, it reduces the time and effort spent on designing crRNAs and primers for the SNV of choice.